Genomewide analysis of the Class III peroxidase gene family in apple (Malus domestica)

Lu Yao 1
Ma Rongqun 2
Wu Kunhao 1
Sun Jilu 2
Li Yutong 1
Zhao Jiawei 1
Qi Zhenbao 3
Sha Guangli 2
Ge Hongjuan ghj042@163.com 2
Shi Yanjing shiyanjing@qust.edu.cn 1
1 College of Biological Engineering, Qingdao University of Science and Technology , Qingdao , China
2 Qingdao Academy of Agricultural Science , Qingdao , China
3 Qilu University of Technology , Jinan , China
Nikalje Ganesh
Electronic publication date: 2025 Aug 18
Publication date: 2025
Volume: 13
Electronic Location ID: e19741
Received 2024 Dec 12; Accepted 2025 Jun 23
Copyright: ©2025 Lu et al.
Copyright year: 2025
Copyright holder: Lu et al.
License: This is an open access article distributed under the terms of the Creative Commons Attribution License, which permits unrestricted use, distribution, reproduction and adaptation in any medium and for any purpose provided that it is properly attributed. For attribution, the original author(s), title, publication source (PeerJ) and either DOI or URL of the article must be cited.
License URL: https://creativecommons.org/licenses/by/4.0/

Keywords: Apple, PRX, Genome-wide, Expression patterns, Rootstock

Funding: China Agriculture Research System, Ministry of Finance and Ministry of Agriculture and Rural Affairs CARS-27 This work is supported by the China Agriculture Research System, Ministry of Finance and Ministry of Agriculture and Rural Affairs (grant number: CARS-27). There was no additional external funding received for this study. The funders had no role in study design, data collection and analysis, decision to publish, or preparation of the manuscript.

==============================
Class III peroxidases (PRXs) play a crucial role in maintaining reactive oxygen species (ROS) homeostasis, thereby influencing plant growth, development, and defense responses. To date, the roles of PRXs in apple branch development and the control of rootstock growth vigor remain poorly understood. This research aimed to exhaustively annotate and analyze the Class III PRX family in the apple genome. Ninety-nine PRX proteins were identified from the Malus domestica genome. Phylogenetic analysis revealed that the PRXs from Malus domestica and Arabidopsis thaliana were classified into six groups. McSCAN analysis indicated that tandem duplication events played a dominant role in the expansion of Malus domestica peroxidases (MdPRXs), thus purifying selection maintained their function. Most MdPRX genes contained cis-elements responsive to light and plant hormones such as abscisic acid (ABA) and methyl jasmonate (MeJA), as well as various stress factors. Although most MdPRXs possess N-terminal signal peptides, in contrast to the majority of Arabidopsis PRX gene family members that are primarily localized in the apoplast, 50 MdPRXs are localized in the chloroplasts, with only one-third predicted to be apoplastic. Analysis of their spatiotemporal expression patterns, based on transcriptome data, revealed extensive involvement in apple tissue and organ development, demonstrating distinct and specialized expression profiles. These variations are primarily attributed to differences in cis-elements within the promoter regions and their three-dimensional structural variations, rather than to their phylogenetic relationships. In rootstock-scion composite trees, the expression patterns of MdPRXs were influenced by both rootstock species and scion varieties. Unlike previous studies relying on zymogram analysis, our findings reveal that the transcriptional expression of MdPRXs is not inherently negatively correlated with the dwarfing capacity of apple rootstocks. Notably, we identified that high expression of MdPRX59 is specifically associated with vigorous rootstocks. A set of MdPRXs such as MdPRX27, MdPRX59, and MdPRX90 may affect the ROS status in stem cell niche of the axillary buds and promote the differentiation of branches. This systematic analysis provides a foundation for the further functional characterization of MdPRX genes, with the aim of improving apple rootstock dwarfing ability and branching characteristics.

Introduction

In the apple industry, the growth vigor of the composite tree (rootstock-scion) and the development of lateral branches are closely related to fruit production. The controlled growth vigor or dwarfing phenotype and more branches differentiation phenotype are desirable to achieve higher productivity as well as early fruiting. Both these two characteristics are regulated by the reactive oxygen species (ROS) continuously formed and removed in plants (Agurla et al., 2018; Dietz, Turkan & Krieger-Liszkay, 2016; Mhamdi & Van Breusegem, 2018; Mittler, 2002; Simmons & Bergmann, 2016; Xu et al., 2017).

Plant peroxidases (PRXs) are critical plant oxidoreductases, catalyzing oxidative reactions in which hydrogen peroxide (H2O2) acts as an electron acceptor. Along with superoxide dismutases (SODs), glutathione reductases (GRs), dehydroascorbate reductases (DHARs), glutathione-S-transferases (GSTs), and glutathione peroxidases (GPXs), PRXs dynamically regulate the generation and removal of reactive oxygen species (ROS) in plants (Jones & Smirnoff, 2005; Mhamdi & Van Breusegem, 2018). Based on their sequences and catalytic characteristics, plant PRXs all contain a heme group composed of protoporphyrin IX and iron (III). There are two classes of PRXs in plants: Class I and Class III peroxidases (Welinder, 1992). Class I PRXs are non-secretory and have few members in plants (Dunand et al., 2011; Li et al., 2015). In contrast, Class III PRXs (Guaiacol PRXs, EC 1.11.1.7), also referred to as secretory PRXs (Hiraga et al., 2000), belong to a multi-gene family, with each species possessing dozens of gene members (Fawal et al., 2012). In this study, we focus on the analysis of Class III PRXs.

Class III PRXs oxidize a wide range of organic and inorganic substrates, including phenolic compounds (e.g., phenolic lignin precursors: coniferyl and sinapyl alcohol), aromatic amines, as well as diamines, indoles (e.g., 3-indole acetic acid, IAA), thiols, and polycyclic aromatic hydrocarbons. Through these redox reactions, they are involved in various physiological processes in plants, including auxin degradation, cell wall modification, lignin formation (Cao et al., 2016; Shigeto & Tsutsumi, 2016), secondary metabolites biosynthesis (Ferreres et al., 2011), and stress response regulation (Kidwai, Ahmad & Chakrabarty, 2020; Li et al., 2020).

In the late 1970s, Cunningham et al. (1975) found that PRX activities were negatively correlated with plant height and internode length in Triticale and Sorghum. Subsequently, dwarfing apple rootstocks were reported to exhibit higher PRX enzyme activities and more PRX isozyme bands (Garcia, Rom & Murphy, 2002). These studies led to the hypothesis that peroxidase (PRX) activity is negatively correlated with the dwarfing ability of apple rootstocks. Specifically, dwarfing rootstocks exhibit higher peroxidase activity, which results in greater oxidation of the auxins synthesized in the roots. This increased oxidation may prevent the effective upward transport of auxins through the phloem, thereby limiting the growth vigor and contributing to the dwarfing phenotype (Lockhard & Schneider, 1981).

Plants, like animals, have stem cells, which are distributed in stem cell niches (SCNs) such as the shoot apical meristem (SAM) and the root apical meristem (RAM) (Sarkar et al., 2007). Cells located in the organizing center (OC) of the SCN divide infrequently and are also known as the quiescent center (QC). These cells are characterized by high superoxide anions (O2•−) levels, low H2O2 levels (Yu et al., 2016; Zeng et al., 2017). Studies have shown that the upregulation of several Arabidopsis PRXs leads to a reduced number of differentiating cells in the peripheral zone of the SAM. Conversely, the number of differentiating cells in the SAM increases with the downregulation of certain AtPRXs (Zeng et al., 2017). Therefore, Class III PRXs may play a crucial role in determining whether an axillary bud differentiates into a branch or remains dormant.

Investigating whether apple PRX genes associated with dwarfing ability of rootstocks, and how they regulate the buds differentiation into branches is of great significance for improving apple production. The low resolution of zymograms and non-specific enzyme activity determination methods employed in earlier biochemical studies have posed challenges in investigating these extensive gene family. However, with the advent of next-generation sequencing, the expression of each specific member of the entire PRX gene family can now be detected and quantified. In recent years, Class III PRX gene families have been identified in perennial woody fruit species, including grapevine (Vitis vinifera) (Xiao et al., 2020), sweet orange (Citrus sinensis) (Li et al., 2020), and pear (Pyrus bretschneideri) (Cao et al., 2016). Their roles in stress responses and fruit development have also been investigated. However, to date, the PRX gene family has not been identified in Malus domestica and its role on apple development remains unknown. In this study, a genome-wide analysis of the Class III PRX gene family in Malus domestica was performed, including phylogenetic relationships, genomic structure, chromosomal localization, and upstream cis-regulatory elements. Multiple RNA-Seq datasets were utilized to analyze their spatiotemporal expression patterns and investigate their expression profiles in different rootstocks exhibiting varying dwarfing abilities, as well as in branching axillary buds. Furthermore, protein 3D structures were predicted to elucidate the antagonistic roles of different apple PRX members. The results provide insights for future studies on the roles of PRX genes in controlling rootstock vigor, apple branch differentiation, and rootstock-scion interactions. This study will also facilitate subsequent gene cloning and functional research on apple Class III PRX members.

Materials & Methods

Gene identification

In the present study, the hidden Markov model (HMM) profile of PRX (PF00141) (Mistry et al., 2021) was downloaded and queried against the apple GDDH13 genome database (https://iris.angers.inra.fr/gddh13/index.html) (Daccord et al., 2017) using HMMER 3.0 software. Default parameters were used with a cutoff value set to 0.001 to identify PRX genes. To verify peroxidase-specific domains, candidate sequences were analyzed using three databases: (1) NCBI’s Conserved Domain Database (CDD) with an E-value threshold of 0.01, (2) the Pfam database (release 35.0), and (3) the SMART database (version 9.0). Sequences lacking a peroxidase-specific domain were excluded from further analysis. Candidate MdPRXs were cross-referenced with RNA-Seq and NCBI EST databases, removing those without detectable expression. The chromosomal locations of all identified PRX members were then mapped using MapChart 2.30 for further analysis.

Phylogenetic analysis

The full length sequences of the characterized PRX proteins were aligned using ClustalW with default parameters. Based on these alignments, phylogenetic trees were constructed in MEGA 7.0, employing the JTT model, pairwise deletion, and 1,000 bootstrap replications. Additionally, for the composite phylogenetic tree, the core peroxidase domain sequences of PRX proteins from Arabidopsis and Malus were aligned, and a neighbor-joining (NJ) tree was constructed using ClustalW in the same manner.

Sequence analysis, gene structure and motif analysis of MdPRXs

The sequence lengths, molecular weights, and isoelectric points of the identified PRX proteins were calculated using TBtools (Chen et al., 2023). Subcellular localization analyses were performed using WoLF PSORT II (https://www.genscript.com/wolf-psort.html). The exon-intron organization of Malus PRX genes was determined by comparing the predicted coding sequences with their corresponding full-length sequences using the Gene Structure Display Server (GSDS: http://gsds.cbi.pku.edu.cn) (Guo et al., 2007). Conserved motifs in MdPRX proteins were identified using MEME Suite version 4.12.0 (Bailey et al., 2009), with the analysis parameters set to detect a maximum of 10 distinct motifs.

Gene duplication

Gene duplication events were analyzed using the Multiple Collinearity Scan toolkit (MCScanX) with default parameter settings (Wang et al., 2012). A Circos plot (Kärkönen et al., 2009) was generated using TBtools software (Chen et al., 2020) to visualize the chromosomal distribution and relationships of segmental duplication gene pairs.

Transcriptome data

MdPRX expression was profiled using RNA-Seq data (GSE253335/276181/274104, PRJNA308148/826123/801073) across rootstocks, tissues, and developmental stages. The detailed RNA-Seq data information was listed in Table 1.

All samples were collected from Qingdao Academy of Agricultural Sciences (36°24′N, 120°58′E) under standard cultivation (fertilization: N in February, nutrient solution in May; pest control: imidacloprid March–May, pyrethroids May–July).

All RNA-seq datasets contain three biological replicates per sample, except for GSE253335, which has two. Samples were sequenced on an Illumina platform with 150 bp paired-end reads. Raw reads were processed with fastp software, to generate clean reads. The clean reads were mapped to the apple genome using HISAT2. The RNA-Seq data were normalized by Fragments Per Kilobase of Exon Per Million Mapped Fragments (FPKM). Differential expression analysis was performed with DESeq2.

RNA extraction and gene expression analysis

50 mg of samples were macro-dissected and immediately flash-frozen in liquid nitrogen, and stored at −80 °C. Total RNA was extracted using the Biospin Plant Total RNA Extraction Kit (Bioer Technology, Hangzhou, China) according to the manufacturer’s instructions. RNA integrity was checked by gel electrophoresis, and concentration was measured with a QuickDrop™ spectrophotometer (Molecular Devices). First-strand cDNA was synthesized using the HiScript III RT SuperMix + gDNA wiper (Vazyme), following the manufacturer’s recommendations. qRT-PCR was performed using the Roche LightCycler® 480 system with SYBR Green I, with apple 18S rRNA as the internal reference. Primers were designed using NCBI Primer-BLAST, targeting the CDS of genes in Malus domestica (amplicons: 100–300 bp; Tm: 60 °C; GC: 40–60%; length: 18–25 bp). Intron-spanning was disabled as gDNA wiper effectively removes gDNA contamination during reverse transcription. Primers with off-target risks were excluded after genome-wide screening. Primers (Table S1) were HPLC-purified (Sangon Biotech). PCR conditions: 95 °C (30 s), then 40 cycles of 95 °C (10 s) and 60 °C (30 s). Reactions were run in triplicates, and gene expression was analyzed by 2−ΔΔCT. Additional data (primer details, PCR specificity) are in File S1. Results (mean ± SD, n = 3) were analyzed with SPSS 18.0 and graphed using ORIGIN 8.0.

Table 1 Summary of RNA-seq datasets analyzed in this study.

Study accession	Cultivar	Tissue/organ	
PRJNA826123	Fuji6	Fuji6 scion graft on WT M. hupehensis (PYTC, vigorous) and its GA mild insensitive mutant A1d (dwarfing rootstock)	
PRJNA801073	WT (M. spectabilis ‘Bly114’) and its multibranching mutant(MB)	Axillary buds	
SRR3095691	M. robusta grafted with WT (M. spectabilis ‘Bly114’) scion and MB scion	Root tip	
GSE274104	M. spectabilis ‘Bly114’	Stems, leaves, flowers	
GSE253335	Dwarfing rootstocks:M9, Budagowski 9 (B9), A1d, M. sylvestris, PYTC	Pholems in active growing stage	
GSE276181	A1d, PYTC	Pholems in bud breaking stage	

Protein structure prediction and comparison

Protein structure prediction of MdPRXs was performed using the Swiss-Model server (https://swissmodel.expasy.org/). The highest-quality models, as determined by GMQE (Global Model Quality Estimation) scores and confirmed as plant class III peroxidases, were downloaded in PDB format and analyzed using PyMOL (version 2.4.0; Schrödinger, LLC) for structural visualization and alignment (Schrödinger, 2021).

Results

Identification of the PRX proteins in apple

The initial HMMER search identified 155 candidate gene models matching the PRX HMM profile (PF00141). PRX domain sequences were extracted from these candidates and aligned using ClustalW, followed by construction of an apple-specific PRX HMM profile using hmmbuild function in HMMER 3.0. A subsequent HMMER search with an E-value cutoff of 0.001 yielded 138 non-redundant candidates. Thirty-nine sequences lacking conserved peroxidase domains in NCBI CDD, Pfam, or SMART databases were excluded. A total of 99 PRX family genes were identified. These MdPRX genes were mapped to the Malus chromosomes based on their physical locations and were renamed MdPRX1 to MdPRX99 according to their order on the 17 apple chromosomes. During the subsequent gene structure and SMART analysis, we identified that MD11G1060900, (renamed as MdPRX61) possessed an unusually long sequence with 19 exons but lacked a signal peptide domain. Furthermore, no peroxidase domain was detected within the first 15 exons. IGV analysis revealed that only the first 15 exons were transcribed, whereas exons 16–19 exhibited no expression in the specific RNA-Seq data, suggesting that MD11G1060900 may consist of two independent transcripts. SoftBerry analysis predicted a novel transcript consisting of the last four exons, which contains both a signal peptide domain and a secreted peroxidase domain. This transcript was manually annotated as MdPRX61 using IGV-GSAman software (Fig. S1).

Physical distribution of MdPRX genes

The original gene IDs and their corresponding renamed IDs, along with their positions on the chromosomes, are available in Table 2. The FASTA information for all MdPRX coding sequences (CDS) is provided in File S2. The lengths of the identified CDS sequences ranged from 489 bp (MdPRX48) to 1,932 bp (MdPRX80), with an average length of 1,000.6 bp. A maximum of 15 MdPRX members were distributed on chromosome 3, followed by chromosomes 13 and 15, with 10 and nine members, respectively. Chromosome 14 contains the fewest MdPRX genes, with only two members, while the other chromosomes harbor between four and eight MdPRX genes (Fig. 1).

Table 2 The MdPRX ID (ID), chromosomal location (CL), proteimn length (Pl), peroxidase domain localization (PDL) data.

MdPRX ID	CL	Pl	PDL	MdPRX ID	CL	Pl	PDL	
MdPRX1	Chr1(11723376-11725311)	337	40–286	MdPRX51	Chr9(7678059-7679599)	326	39–287	
MdPRX2	Chr1(11727415-11729067)	327	33–282	MdPRX52	Chr9(7682031-7684010)	327	43–290	
MdPRX3	Chr1(11802786-11804049)	328	33–274	MdPRX53	Chr10(4311879-4315696)	349	50–293	
MdPRX4	Chr1(26715130-26716536)	342	65–306	MdPRX54	Chr10(30519595-30522348)	397	82–357	
MdPRX5	Chr1(26724247-26725944)	318	41–282	MdPRX55	Chr10(31634635-31636205)	347	67–307	
MdPRX6	Chr1(26756983-26758929)	319	41–283	MdPRX56	Chr10(32294335-32296319)	356	78–317	
MdPRX7	Chr1(29266670-29268274)	359	46–281	MdPRX57	Chr11(40317552-40320129)	337	53–297	
MdPRX8	Chr1(29285647-29287508)	328	45–293	MdPRX58	Chr11(351846-354820)	358	83–322	
MdPRX9	Chr2(10148695-10150363)	318	37–286	MdPRX59	Chr11(1212949-1214635)	324	42–288	
MdPRX10	Chr2(11603248-11605294)	331	46–294	MdPRX60	Chr11(1220317-1222110)	332	46–295	
MdPRX11	Chr3(978704-981129)	300	1–250	MdPRX61	Chr11(5307350-5308724)	319	42–283	
MdPRX12	Chr3(1007836-1010169)	309	47–280	MdPRX62	Chr11(5310626-5312104)	320	42–284	
MdPRX13	Chr3(1039521-1041488)	351	48–301	MdPRX63	Chr11(5324074-5325955)	322	44–286	
MdPRX14	Chr3(1071235-1073468)	351	48–301	MdPRX64	Chr11(5338009-5341504)	327	48–291	
MdPRX15	Chr3(1100985-1104235)	262	51–262	MdPRX65	Chr11(35095555-35099206)	315	42–279	
MdPRX16	Chr3(1111772-1113496)	324	42–288	MdPRX66	Chr12(26519131-26521745)	323	47–287	
MdPRX17	Chr3(1145360-1147296)	324	42–288	MdPRX67	Chr12(26549366-26554066)	223	1–223	
MdPRX18	Chr3(1155773-1158026)	331	46–295	MdPRX68	Chr12(28198055-28199212)	327	44–290	
MdPRX19	Chr3(4745447-4749224)	383	106–347	MdPRX69	Chr13(666151-668317)	327	49–291	
MdPRX20	Chr3(4750789-4752469)	330	50–294	MdPRX70	Chr13(677492-678650)	328	49–292	
MdPRX21	Chr3(4774413-4776918)	349	70–313	MdPRX71	Chr13(3757449-3759212)	338	49–300	
MdPRX22	Chr3(6813999-6814865)	162	1–122	MdPRX72	Chr13(6068702-6069789)	333	41–293	
MdPRX23	Chr3(16353084-16354911)	336	47–290	MdPRX73	Chr13(6930968-6932144)	339	53–302	
MdPRX24	Chr3(30818821-30821801)	315	42–279	MdPRX74	Chr13(6935956-6937164)	339	53–302	
MdPRX25	Chr3(35765975-35767900)	324	40–286	MdPRX75	Chr13(11797366-11798748)	335	54–299	
MdPRX26	Chr4(5396644-5399902)	341	57–303	MdPRX76	Chr13(11978084-11979797)	324	36–288	
MdPRX27	Chr4(18814701-18817577)	328	47–290	MdPRX77	Chr13(14485822-14487459)	310	43–277	
MdPRX28	Chr4(26248958-26251656)	324	48–288	MdPRX78	Chr13(25989270-25991264)	326	41–290	
MdPRX29	Chr4(29623037-29624583)	305	42–269	MdPRX79	Chr14(989088-990715)	322	38–280	
MdPRX30	Chr5(15957716-15959760)	333	50–297	MdPRX80	Chr14(27785079-27789211)	643	33–285; 350–603	
MdPRX31	Chr5(35712401-35714983)	400	84–360	MdPRX81	Chr15(1283048-1285216)	361	76–325	
MdPRX32	Chr5(37634025-37635795)	356	78–317	MdPRX82	Chr15(7552935-7555121)	345	56–309	
MdPRX33	Chr5(43863293-43865116)	353	64–315	MdPRX83	Chr15(13460515-13462105)	321	38–280	
MdPRX34	Chr5(46533897-46536242)	335	51–295	MdPRX84	Chr15(19672431-19674217)	318	37–286	
MdPRX35	Chr6(4979437-4982303)	415	124–378	MdPRX85	Chr15(21319998-21322083)	358	73–321	
MdPRX36	Chr6(32013448-32014425)	325	33–285	MdPRX86	Chr15(24886316-24888457)	328	42–289	
MdPRX37	Chr6(32016607-32017584)	325	32–285	MdPRX87	Chr15(27448281-27450997)	314	28–278	
MdPRX38	Chr7(30489128-30490950)	336	60–300	MdPRX88	Chr15(33504699-33506002)	337	33–283	
MdPRX39	Chr7(30507239-30509107)	319	41–283	MdPRX89	Chr15(33511841-33513542)	360	70–316	
MdPRX40	Chr7(30512492-30513857)	329	47–290	MdPRX90	Chr16(3719783-3721576)	341	50–303	
MdPRX41	Chr7(36009874-36011393)	331	45–293	MdPRX91	Chr16(5990099-5991264)	361	41–294	
MdPRX42	Chr8(2023966-2025791)	322	58–286	MdPRX92	Chr16(6891481-6892745)	340	50–304	
MdPRX43	Chr8(4054705-4057058)	329	47–292	MdPRX93	Chr16(6894709-6895929)	336	50–301	
MdPRX44	Chr8(28531555-28533332)	321	38–279	MdPRX94	Chr16(12153811-12155544)	324	36–288	
MdPRX45	Chr8(29618758-29619968)	327	44–286	MdPRX95	Chr16(15066943-15068633)	327	45–287	
MdPRX46	Chr8(29625015-29626302)	333	48–292	MdPRX96	Chr17(2928948-2930648)	327	37–290	
MdPRX47	Chr8(29632630-29633745)	331	48–290	MdPRX97	Chr17(7730960-7732680)	325	38–286	
MdPRX48	Chr9(676891-677637)	219	1–182	MdPRX98	Chr17(7841205-7842915)	326	41–287	
MdPRX49	Chr9(679939-681272)	341	53–304	MdPRX99	Chr17(32582249-32583484)	314	36–278	
MdPRX50	Chr9(2388224-2389207)	327	37–290					

Phylogenetic relationships of MdPRX proteins in apple and Arabidopsis

To investigate the evolutionary relationships, we used the PRX domain amino acid sequences of 99 MdPRXs and 73 AtPRXs from Arabidopsis thaliana to construct a neighbor-joining (N-J) tree using MEGA 7.0. These PRX proteins were clustered into six groups. According to the phylogenetic distances, Group 1 was further subdivided into four subgroups (A, B, C, and D), and it represented the largest group, comprising 73 PRX members (46 from apple and 27 from Arabidopsis). Group 4 ranked as the second-largest cluster, containing 35 PRX proteins (18 from apple and 17 from Arabidopsis). Group 4 consisted of 26 PRX proteins, with 11 originating from Arabidopsis and 15 from apple. In contrast, Group 2, 3, and 6 were the smallest groups, each containing only 13 PRX members (Fig. 2).

Figure 1 The chromosomal localization of MdPRX genes in Malus domestica genome generated by MapChart 2.30.

Figure 2 Phylogenetic relationship of PRXs from apple and Arabidopsis.

The PRX domain sequences of Arabidopsis and Malus were aligned, and a neighbor-joining (N-J) tree was constructed using MEGA 7.0 with 1,000 bootstrap replications. The numbers next to the branches indicate the bootstrap values, which represent the evolutionary proximity and reliability of the inferred phylogenetic relationships between the genes. Higher bootstrap values suggest stronger support for a particular branch, indicating greater confidence in the evolutionary grouping of the sequences.

Absolutely purifying selection for MdPRX genes

To elucidate the evolutionary expansion of MdPRX gene family, McScanX was employed to analyze duplication events of MdPRX genes. The results showed that 56 MdPRX genes were involved in duplication events, resulting in 38 pairs of segmental duplications among MdPRX members. Additionally, seven pairs of segmental duplications were identified between MdPRX members and other regions of the apple genome. No tandem repeat events were detected, suggesting that segmental duplication plays a dominant role in the expansion of the MdPRX gene family. A Circos map illustrating these duplication events was generated based on collinearity analysis. In apple, Chr 03 and Chr 11, Chr 13 and Chr 16, and Chr 05 and Chr 10 are copies of the same ancestral chromosome (Velasco et al., 2010). In this study, we observed strong collinearity of MdPRX members between Chr 13 and Chr 16, Chr 03 and Chr 11, and Chr 05 and Chr 10, whereas the MdPRX members on Chr 08 exhibited no collinearity with other chromosomes (Fig. 3).

Figure 3 Circos map the MdPRX genes.

The rings indicate (from outside to inside), heat maps representing gene density, blue dots representing N ratio and green lines representing GC ratios of each chromosomes (Chr). The sizes of each chromosome are shown on the scale. Numbers along each chromosome box indicate sequence lengths in Mb. A map connecting homologous regions of the apple genome is shown inside the figure. The red lines indicated the collinear relationship between the MdPRX genes, and the gray line indicated the collinear relationship between genes of the whole apple genome.

The selection pressure among different types of duplication was also inferred by calculating the rates of synonymous substitution (Ks) and non-synonymous substitution (Ka). During evolution, genes typically undergo purifying selection (Ka/Ks < 1), positive selection (Ka/Ks > 1), or neutral selection (Ka/Ks = 1) (Khan et al., 2019). In this study, most of the Ka/Ks values for the MdPRX collinear pairs were less than 0.5, excerpt for two pairs (MdPRX27 vs MD12G1121300 and MdPRX77 vs MdPRX91) showed values around 0.5–0.6 (Fig. 4, File S3). The divergence time of each pair ranged from 3.61 million years ago (Mya) to 116.28 Mya. Therefore, we propose that the MdPRX gene family has undergone strong purifying selection at a slow evolutionary rate.

Figure 4 Scatter plots of the Ka/Ks ratios of duplicated PRX genes in apple.

X-axis represents the synonymous distance: Ks, and y-axis the Ka/Ks ratio for each pair, respectively.

The first class III peroxidase appeared approximately 450 Mya, coinciding with the emergence of terrestrial plants. Since then, the number of gene copies has increased significantly. This expansion appears to be correlated with the evolution of plant architecture and complexity, as well as the diversification of biotopes and pathogens (Passardi et al., 2005). The number of MdPRXs (99) exceeds that of Arabidopsis thaliana (73), grapevine (47), and pear (94). This difference stems from to apple’s more complex plant architecture in contrast to the herbaceous nature of Arabidopsis thaliana and the vine-like structure of grapevine. Notably, both pear and apple evolved from a common ancestor that existed around 5–20 Mya. However, apple possesses a greater number of PRX genes than pear, which likely contributes to its superior adaptability—particularly in terms of cold and drought tolerance—as well as its broader ecological adaptability. This phenomenon of gene family expansion due to enhanced adaptability is also observed in the WRKY gene family, with apple possessing 118 members compared to pear’s 103 (Huang et al., 2015; Lui et al., 2017).

Gene characteristics and cellular sublocallization of MdPRXs

Gene characteristics, including protein sequence length, molecular weight (MW), isoelectric point (pI), and subcellular localization, were analyzed (File S4). Protein lengths ranged from 162 aa (MdPRX22) to 643 aa (MdPRX80), with an average length of 332.5 aa. The MW ranged from 17,702.1 Da (MdPRX22) to 70,893.1 Da (MdPRX80), with an average of 36,268.7 Da. The isoelectric point (pI) ranged from 4.03 (MdPRX12) to 9.63 (MdPRX41), with an average of 7.40. Most MdPRXs (84 out of 99) exhibited hydrophobicity. According to TargetP (https://services.healthtech.dtu.dk/services/TargetP-2.0/), 88 out of 99 MdPRX members possess a signal peptide sequence, whereas 11 MdPRXs lack signal peptide sequences (File S5). According to Wolf PSORTII predictions, two MdPRXs (MdPRX29 and MdPRX48) of them were predicted to localize to the cytoplasm, whereas the other nine MdPRXs were predicted to be chloroplast-localized. 33 MdPRXs were predicted to localize to extracellular regions, while 50 (including the nine members without signal peptides) and one member were predicted to localize to chloroplasts and mitochondria, respectively. Notably, 10 MdPRXs were predicted to localize to vacuoles, however, all of these vacuole-localized PRXs possess signal peptides. The remaining MdPRXs were predicted to localize to the endoplasmic reticulum (ER), plasma membrane, and mitochondria (File S4).

Gene structure, motif and conserved domain analysis of MdPRXs

The 10 conserved motifs (motifs 1-10) of  MdPRX were identified using the MEME program. Notably, Motif 1, Motif 10, Motif 2, and Motif 9 were the most common among MdPRXs (Fig. 5B). LOGOs for these motifs were also generated from MEME (Fig. S2). The highest consensus sequence (41) was observed in Motif 1, while the lowest (11) was recorded in Motif 9. All MdPRX members, except for MdPRX11, MdPRX19, MdPRX22, MdPRX48, MdPRX67, MdPRX87 and MdPRX80, contain all ten conserved motifs arranged in the following order: motif 1, motif 10, motif 2, motif 9, motif 5, motif 8, motif 3, motif 4, motif 7, and motif 6. Notably, MdPRX80 featured two sets of motifs (Motifs 1 to 5 and Motifs 7 to 10), but contained only a single copy of Motif 6 (Fig. 5A).

Figure 5 Phylogenetic relationships, gene structure and architecture of conserved protein motifs in MdPRXs.

(A) Unrooted neighbour-joining phylogeny of MdPRXs based on full length protein sequences of MdPRXs, with bootstrap values = 1,000. (B) The motif composition of MdPRX proteins pridicted by MEME-v4.12.0 software, a color-coded legend representing motif 1-10 is displayed on the right panel.The sequence information for each motif is provided in Fig. S2, the length of protein can be estimated using the scale at the bottom. (C) The schematic integrates the gene intron-exon structure with SMART domain architecture predictions, where black lines denote introns, light green boxes indicate 5′/3′ UTRs, yellow boxes represent exons, and color-coded functional domains are displayed as follows: pink for signal peptides, and dark green for Pfam peroxidase domains (potentially fragmented by intervening introns). Additionally, those marked with an asterisk (*) do not contain the signal peptide domain predicted byTargetP. Relative protein or gene lengths can be estimated by gray bars.

Gene structure analysis revealed that MdPRX members had between one and four exons, with most containing four. Diverging from high structural conservation observed in protein architecture across the whole gene family, MdPRX gene structures exhibited partial phylogenetic coherence. Paralogous clusters within subclades maintained conserved exon-intron architectures, for example:six genes (MdPRX50/96/72/91/36/37) which clustered within the same subclade, all contained only one exon (Fig. 5C).

Batch SMART analysis revealed that all MdPRXs, contain a single Pfam peroxidase domain, except for MdPRX80, which contains two distinct Pfam peroxidase domains (Table 2). In all cases, the peroxidase domains encompass all exons (Fig. 5C). The majority of MdPRXs contain a signal peptide domain at the N-terminus (Fig. 5C; File S6).

Cis-regulatory element analysis of MdPRXs

The analysis of cis-regulatory elements within the 2,000 bp upstream regions of the sequences is presented. MdPRXs clustered within the same phylogenetic clades exhibit distinct cis-element profiles in their upstream regulatory regions, suggesting divergent regulatory mechanisms and transcriptional responsiveness (Fig. 6).

Figure 6 The cis-elements of predicted MdPRXs in upstream 2,000 bp regions.

The phylogenetic tree of MdPRXs (constructed based on their full-length protein sequences) is shown on the left panel, while a color-coded legend representing distinct cis-element categories is displayed on the right panel, the −2,000 bp upstream sequence length scale bar is underneath.

Among the identified cis-elements of MdPRXs, light-responsive elements constituted 61% (Table S2), suggesting a potential role for MdPRXs in photoprotection through ROS scavenging. O2•− could be produced during photosynthesis in the chloroplasts, but their accumulation can lead to oxidative stress, damaging various cellular components (Asada, 1999). Once O2•− is converted to H2O2 by SOD, CATs and PRXs further degrade H2O2 into water and oxygen, thereby completing the detoxification process. This step is essential because H2O2, although less reactive than O2•−, can still cause oxidative damage if not removed (Zeng et al., 2017).

Approximately 13% of identified cis-elements were methyl MeJA-responsive, 5% drought-responsive, and 4% ABA-responsive elements. These findings align with previous studies on the enzymatic and molecular functions of PRXs, providing additional evidence for their dual roles in biotic and abiotic stress responses (New, Barsky & Uhde-Stone, 2023). Additionally, 4% and 2% of the identified cis-elements are associated with gibberellin- and auxin-responsive elements, respectively, suggesting that these PRXs may be involved in growth regulation. Elements specific to meristem, endosperm, and seed tissue expression, as well as those involved in circadian control, were also identified within the promoters of MdPRXs.

Spatio-temporal expression patterns of MdPRXs

The average FPKM value from three replicates for each MdPRX was obtained from transcriptomic data of axillary buds, leaves, stems (young shoots), and flowers of Malus spectabilis ‘Bly114’ grafted onto M. robusta, as well as from M. robusta root tips (GSE274104 and PRJNA801073). A total of 68–89 members were expressed in these tissues, with a minimum of 68 in buds and a maximum of 87 in roots, indicating a preferential accumulation of MdPRXs in roots. More than half (55 out of 99) of MdPRXs exhibited constitutive expression patterns, while some members showed strict tissue-specific expression patterns. For example, MdPRX62, MdPRX63, MdPRX22, and MdPRX30 were exclusively expressed in roots. Others exhibited significantly elevated expression levels (log2 fold change >2) in particular tissues relative to others, as evidenced by their distinct red coloration in the heatmap (data were log2-transformed and subjected to row normalization). For example, MdPRX 38/83/95/99 exhibited predominant expression in stem tissues, whereas MdPRX4 and MdPRX97 displayed leaf-preferential expression patterns (Fig. 7A).

Figure 7 The expression profiles of MdPRXs.

Based on the phylogenetic relationships shown in Fig. 5, the genes were subdivided into Groups 1–7, which are indicated by colored boxes in green, blue, yellow, orange, red, gray, and purple, respectively. (A) Spatio-temporal expression patterns of MdPRXs. (B) Expression patterns of MdPRXs in wild-type (WT) and more-branching (MB) axillary buds during side shoots outgrowth. (C) Expression analysis of MdPRX87 by qRT-PCR, with data normalized to the 18S rRNA gene; vertical bars indicate standard deviation. (D) The expression patterns of MdPRXs in A1d (a GA partially insensitive mutant of WT), PYTC(WT M. hupehensis), B9, M9, and M. sylvestris during active growth stages, and in A1d0 and PYTC0 (A1d and PYTC at the bud break stage). (E) The agrose gel electrophoresis of the RT-PCR of MdPRX59, Lane M:DL2000 marker, lane 1–8 represent cDNA of phloem of A1d, B9, M9, M. hupehensis, M. sylvestris, QZ1, M. sieversii and M. baccata. (F) The expression profiles of MdPRXs in Fuji6 scions grafted onto A1d (AF) and WT M. hupehensis (PF). (G) The expression profiles of MdPRXs in the root tip of M. robusta rootstock grafted with scions of WT apple (M. spectabilis) and a more-branching (MB) mutant. The heat maps were generated using TBtools software, based on relative expression levels of MdPRX genes, the row normalized log2 transformed values were used with hierarchical clustering.

Interestingly, the expression of MdPRX15 was not detected in any of the tissues analyzed in this study. However, a BLAST search of its CDS against the NCBI EST database revealed that MdPRX15 is expressed in xylem tissue of Royal Gala plants subjected to 5 °C for 24 h at a specific developmental stage (EST: lIBEST_024520).

The widespread distribution of MdPRX s across various non-stressed apple tissues and organs (with the exception of MdPRX15) strongly supports their multifunctional roles in developmental regulation, particularly through tissue-specific specialization of individual members. This conclusion is consistent with and substantiated by previous studies in model systems, including Arabidopsis leaf, stem, and root development, as well as transgenic analyses in both tobacco and apple systems. In Arabidopsis thaliana, a set of PRXs has been implicated in regulating leaf cell expansion and determining final organ size by modulating ROS homeostasis in the apoplas (Lu et al., 2014). In stems, peroxidases contribute to lignin deposition. They play a role in xylem differentiation and vascular tissue development (Hoffmann et al., 2020). Arabidopsis PRX8 transgenic apple showed an increase in the number of xylem vessels in the stem (Vicuna, 2005). Arabidopsis PRX01, PRX44, and PRX73 participate in root hair formation and root tissue differentiation (Marzol et al., 2022). Based on microarray analysis, AtPrx13, AtPrx30 and AtPrx55 potentially involved in Arabidopsis flower and fruit development (Cosio & Dunand, 2010).

The expression patterns of MdPRXs during axillary bud sprouting

Quiescent multibranching mutant (MB) axillary buds, minimally branching buds, and elongation zones of new branches were designated as MB1, MB2, and MB3, respectively (Fig. S3). Meanwhile, WT (Malus spectabilis ‘Bly114’) axillary buds lacking branching potential were designated as WB. RNA-Seq analysis was performed on MB1, MB2, and WT axillary buds (PRJNA801073). As illustrated in Fig. 7B, Seventy-five MdPRXs were expressed in these three types of buds and clustered into five clades. MdPRXs in Clade 1 were downregulated in MB2, while those in Clade 2 were upregulated in MB2, indicating their involvement in the axillary bud-breaking process. MdPRXs belonging to Clade 3 exhibited higher expression in WB but lower expression in MB1 and MB2, suggesting their role in maintaining the stemness of WB.

Traditionally, PRXs are considered H2O2 consumers, transferring electrons from other substrates to H2O2 and reducing it to water. The function of Clade 3 MdPRXs aligns with this traditional role. However, Class III PRXs participate in three distinct cycles—peroxidative, oxidative, and hydroxylic (Abbas et al., 2018). While they reduce H2O2 to water in the peroxidative cycle, they also produce hydroxyl radicals (OH•) in the hydroxylic cycle and superoxide anions (O2•−) in the oxidative cycle (Passardi et al., 2005; Shigeto & Tsutsumi, 2016). Thus, PRXs function as bifunctional reductases, both consuming H2O2 and generating ROS species such as OH• and O2•−. In this study, Clade 4 and 5 PRXs are upregulated in MB2 and/or MB1 (Fig. 7B). Buds must maintain high O2•− levels and low H2O2 levels to preserve stemness. However, to break the bud dormancy, H2O2 should accumulated in the peripheral (elongation) zone to inhibit WUS (WUSCHEL), a key regulator of stem cell fate in plant meristems, thus differentiation take place (Zeng et al., 2017). Given that elevated H2O2 concentrations in the peripheral zone are associated with differentiation, these upregulated MdPRXs may contribute to H2O2 production rather than its elimination.

The expression profile of MdPRX87, a Clade 3 member, was analyzed via qRT-PCR, revealing a gradual decrease across the SCN region in MB1 (quiescent lateral bud), MB2 (sprouting axillary bud), and the elongation zone in MB3 (developed lateral shoot). The highest expression level was observed in WT axillary buds, which are incapable of branching (Fig. 7C). This suggests that MdPRX87 may contribute to H2O2 elimination in the QC region of axillary buds. As its expression declines, H2O2 concentration increases in the peripheral zone of MB buds, promoting lateral branch differentiation. In contrast, the high expression level of MdPRX87 in WT axillary buds may suppress differentiation of the lateral branches.

The expression patterns of MdPRXs in dwarfing and vigorous rootstocks across different growth stages

A total of 78 MdPRXs were expressed in phloem tissues. In RNA-seq dataset GSE253335, 45, 50, 69, 45, and 54 MdPRXs were detected in the actively growing phloem of A1d, Budagowski 9 (B9), M. sylvestris, M. hupehensis (PYTC), and M9, respectively. Meanwhile, 32 and 49 MdPRXs were expressed in the bud-breaking phloem of A1d (A1d0) and M. hupehensis (PYTC0) (RNA-seq dataset GSE276181), respectively. Overall, more MdPRXs were expressed during active growth than at the bud-breaking stage. Remarkably, five MdPRX genes—MdPRX1, MdPRX28, MdPRX50, MdPRX57, and MdPRX61—exhibited constitutive high-level expression patterns across all examined growth stages in both vigorous and dwarfing rootstock genotypes. In contrast, MdPRX79, MdPRX96, MdPRX58, MdPRX39, and MdPRX53 were also ubiquitously expressed but at lower levels. MdPRX5, MdPRX35, MdPRX71, and MdPRX7 were highly expressed in the actively growing phloem of both vigorous and dwarfing rootstocks; however, they exhibited little to no expression at the bud-breaking stage (Fig. 7D).

MdPRX99 is highly expressed in the dwarfing rootstock M26, in contrast, MdPRX59 was exclusively expressed in vigorous rootstocks (M. hupehensis and M. sylvestris) across both bud-breaking and actively growing stages. Further RT-PCR analysis of MdPRX59 in the phloem of four dwarfing rootstocks (A1d, B9, M9, QZ1) and four vigorous rootstocks (M. hupehensis, M. sylvestris, M. sieversii, M. baccata) at the active growth stage (Fig. 7E) confirmed that MdPRX59 was expressed exclusively in vigorous rootstocks. These findings posit MdPRX59 expression deficiency as both a predictive biomarker for dwarfing rootstock selection and a candidate breeding target. Mechanistically, targeted downregulation of MdPRX59—achievable through CRISPR/Cas9-mediated knockout or RNAi silencing—represents a strategy for engineering rootstocks with elevated dwarfing ability, warranting prioritized functional validation in apple breeding programs. Concurrently, complementary investigations should evaluate whether MdPRX99 over-expression conversely enhances dwarfing capacity.

Based on total FPKM values, the previous proposal that higher PRX activity correlates with dwarfing ability was supported by comparisons between WT M. hupehensis (PYTC) and its GA-insensitive dwarf mutant (A1d), as well as between M9 and the two types of vigorous rootstocks. However, this correlation did not hold when comparing B9 with vigorous rootstocks. B9 exhibited the lowest total PRX transcription level, however it is known for its strong dwarfing ability among several rootstocks (e.g., M.9T 984, M.9T 337, Jork 9, Mark 9, B9, M.9 EMLA, Pajam 1, Pajam 2, and Supporter 4) (Gjamovski & Kiprijanovski, 2011). In addition, there was little difference between total PRX transcription of the vigorous M. sylvestris and the dwarfing rootstock A1d.

Previous studies have demonstrated that PRX activity progressively increases during plant development, correlating with the accumulation of lignin content. In accorded with this proposal, we found that the total of PRX transcripts of A1d at the bud-breaking stage was much lower than that of at actively growing stage. However, in the vigorous rootstock M. hupehensis, higher total PRX expression was observed at bud-breaking stage than that of at actively growing stage.

These discrepancies suggest that PRXs may have bifunctional or antagonistic roles in regulating plant growth vigor, an assertion supported by existing studies. Different PRX members exhibit opposing functions in cell wall modification, as well as in auxin (IAA) metabolism and signaling (Chen & Schopfer, 1999; Francoz et al., 2015; Kawano, 2003; Schopfer et al., 2002). Some PRXs oxidize aromatic cell wall compounds in the presence of H2O2, thereby stiffening the cell wall and causing growth retardation (Francoz et al., 2015; Passardi, Penel & Dunand, 2004). In contrast, other PRX members may promote cell division and growth by reducing local H2O2 concentrations or generating OH• that break covalent bonds in cell wall polymers (Schopfer, 2001). In relation to IAA metabolism and signaling, PRX-mediated signaling primarily facilitates the degradation (oxidation) of both IAA and its downstream signaling molecule, H2O2, through the conventional PRX cycle. Meanwhile, extracellular IAA activates a plasma membrane-bound NADH oxidase, stimulating the apoplastic production of superoxide anions (O2•−), a portion of which is subsequently converted into H2O2. The generated H2O2 then serves as a substrate for PRX enzymes bound to cell wall polymers, leading to the production of hydroxyl radicals (OH•). These OH• radicals cleave the backbone of cell wall polysaccharides, thereby mediating IAA-induced cell elongation and ultimately promoting plant growth (Chen & Schopfer, 1999; Kawano, 2003; Schopfer et al., 2002).

Rootstock-Scion interaction on MdPRX expression characteristics

The expression profiles of MdPRXs in Fuji6 scions grafted onto vigorous rootstock WT M. hupehensis (PF) and its GA-insensitive dwarfing mutant A1d (AF) were analyzed using RNA-Seq data (PRJNA826123). The results indicated that rootstocks influence the expression of MdPRXs in scions. Specifically, MdPRX26, MdPRX9, MdPRX5, MdPRX4, MdPRX95, and MdPRX38 were significantly downregulated in AF compared to PF, whereas MdPRX35 exhibited significant upregulation in AF. Additionally, the total MdPRX transcription level, calculated as the sum of FPKM values, was lower in AF than in PF (Fig. 7F).

Notably, our previous study documented a statistically significant reduction (p < 0.05, Student’s t-test) in internode length between AF and primary PF (Shi et al., 2023). This observation aligns with reports indicating that Norway spruce plants overexpressing the peroxidase-like gene spi2 exhibited higher guaiacol peroxidase activity concomitant with reduced height and cell length (Elfstrand et al., 2001). Despite these precedents, the correlation between altered MdPRX expression and scion performance still needs to be validated through transgenic experiments.

It was also observed that the root tips of M. robusta rootstocks grafted with scions from WT M. spectabilis and a more-branching (MB) mutant exhibited distinct MdPRX expression patterns (Fig. 7G). Additionally, the primary root length, diameter, and root weight of rootstocks grafted with MB scions were significantly reduced compared to those grafted with WT scions (Li et al., 2016). Thirty-one PRX members were significantly upregulated, while six were downregulated in the root tips grafted with MB scions compared to those grafted with WT scions. This suggests that the interaction between scions and rootstocks influences MdPRX expression patterns, which in turn affects root growth and development. The result is consistent with findings that overexpression of a tobacco anionic peroxidase gene led to a 50% reduction in root weight in transgenic tobacco plants (Lagrimini et al., 1997). It also consistent with the study proposed that the accumulation of peroxidase-related ROS in trichoblasts correlates with root hair initiation in barley (Kwasniewski et al., 2013). These results suggest that scions may regulate MdPRX expression in the RAM, thereby influencing the differentiation of rootstock root tips.

MdPRX protein structures

The three-dimensional structures of six representative MdPRX proteins were predicted using homology modeling based on high-quality templates retrieved from the Protein Data Bank (PDB) (Berman et al., 2000) or from the AlphaFold Protein Structure Database (https://alphafold.ebi.ac.uk; DeepMind and EMBL-EBI, CC-BY 4.0) (Jumper et al., 2021). The AlphaFold templates (AFDB IDs M5XQS7, M5XC68, A0A2P6RAP4, and A0A834ZTL9, corresponding to MdPRX76, MdPRX90, MdPRX27, and MdPRX99, respectively) and the PDB templates (PDB IDs 4cuo and 1qgj, corresponding to MdPRX59 and MdPRX2, respectively) were utilized, with all models meeting a stringent Global Model Quality Estimation (GMQE) threshold of ≥0.85 to ensure structural reliability. The results revealed that MdPRXs exhibit highly similar structural architectures. As illustrated in Fig. 8A, six MdPRXs homologous structures were aligned with near-perfect precision. These proteins are characterized by two predominantly α-helical domains, along with a smaller third domain, the “β-domain”. Collectively, these three structural domains constitute a conserved pocket-like architecture at their interface. The heme ligand is anchored to the PRXs via a network of both polar and nonpolar interactions within the pocket-like interdomain configuration. This arrangement not only stabilizes the heme but also creates an accessible environment for substrates to enter and engage with the heme, thereby, facilitating their catalytic function. Notably, the size of these pockets varies among different MdPRX members, with some displaying larger pockets compared to others. Specifically, MdPRX2, MdPRX76, and MdPRX99 exhibit larger pockets, as shown in Fig. 8B, while MdPRX27, MdPRX59, and MdPRX90 feature smaller pockets, as depicted in Fig. 8C. Studies suggest that PRX-oxidized compounds (e.g., lignin precursors) require heme-edge accessibility to undergo catalytic transformation (Kwon, Moody & Raven, 2015). This suggests that variations in pocket size among different PRXs may influence their substrate accessibility and catalytic efficiency, thereby modulating their biological roles. Our results are align with the proposal derived from AlphaFold modeling (New, Barsky & Uhde-Stone, 2023) that PRXs capable of producing ROS exhibit distinct interdomain and protein–heme interactions, potentially functioning as ‘gatekeepers’ by preventing larger substrates from accessing the heme.

Figure 8 The predicted 3D structures of six MdPRXs.

(A) Views of six templates alignment illustrate the bound heme positioned between the distal domain, proximal domain, and the beta domain. AlphaFold templates: M5XQS7 (red), M5XC68 (gray), A0A2P6RAP4 (orange), and A0A834ZTL9 (cyan) correspond to the homologs MdPRX76, MdPRX90, MdPRX27, and MdPRX99, respectively. Additionally, the PDB templates four cuo (rose red) and one qgj (blue) correspond to the homologs MdPRX59 and MdPRX2, respectively. The left panel shows a side view, while the right panel represents a front view. (B) Surface views of the homologs MdPRX2, MdPRX76, and MdPRX99 are presented, respectively, illustrating larger substrate-binding pockets. (C) Surface viwes of MdPRX27, MdPRX59, and MdPRX90 homologs are presented, respectively, illustrating smaller substrate-binding pockets.

In the context of axillary bud sprouting, MdPRX59 /27/90, which are expressed at higher levels in the sprouting MB2 compared to the quiescent MB1 and WB, feature a smaller heme-containing interdomain. This structural characteristic may enhance their capacity for ROS production, leading to an increase in local H2O2 concentration within the peripheral zone of the SCN. Such a shift could disrupt the delicate balance of high O2•− and low H2O2 levels in the SCN, thereby promoting differentiation and initiating the sprouting process. Conversely, MdPRX2/76, which have larger heme-substrate pockets, may act as ROS scavengers, thereby reducing the concentration of H2O2 in the SCN. Notably, these genes demonstrated the highest expression levels in WB, while displaying relatively lower expression in both MB1 and MB2, implying their potential role in maintaining low H2O2 concentrations in the dormant WB by consumping it.

Concerning the regulation of rootstock growth vigor, MdPRX99, which exhibited the highest expression in the dwarfing rootstock M26, is classified as a large-pocket PRX. It may function as a conventional PRX, oxidizing larger substrates such as IAA and lignin precursors, thereby contributing to growth inhibition. In contrast, MdPRX59, which is specific to vigorous rootstocks, has been identified as a smaller-pocket PRX. It may play a role in the production of ROS, specifically OH• and O2•−, thereby promoting growth by modulating cell wall elasticity.

Discussion

Purifying selection versus cis-regulatory divergence drive functional diversification

Phylogenetic analysis of MdPRXs coupled with McScanX-based genomic collinearity assessment revealed distinct evolutionary trajectories of gene duplication events. Notably, most segmentally duplicated gene pairs were clustered within the same phylogenetic subclades and exhibited minimal genetic divergence, suggesting functional conservation during evolution. Only a limited subset of duplication pairs exhibited sequence divergence after duplication. Subsequent evolutionary rate analysis (Ka/Ks < 1) further confirmed that MdPRXs evolved under strong purifying selection. These findings were supported by conserved structural patterns in gene motif architecture, where paralogs across divergent phylogenetic groups retained nearly identical motif organization and composition profiles, indicating strong protein level sequence conservation. Beside the segmental duplication gene pairs, several groups of consecutive MdPRX members were observed in the same branches of phylogenetic tree, such as MdPRX11-15 in Group 1a and MdPRX45-47 in Group 2 formed tight phylogenetic clusters (Fig. 3). These consecutive MdPRXs were not classified as tandem duplication events due to McScanX’s inherent filtering parameters. McScanX employs a consolidation protocol for collinear gene identification wherein consecutive BLASTP matches sharing a common anchor gene are algorithmically processed. Specifically, if paired homologs reside within a genomic interval of fewer than five intervening genes, these matches are computationally merged, retaining only the representative gene pair exhibiting the most significant sequence similarity (i.e., the lowest BLASTP E-value). This filtering mechanism prevents overcounting of tightly linked homologs (Wang et al., 2012). These local tandem arrays are evolved from very early duplications in the phylogenies, for which mechanisms are difficult to infer (Cannon et al., 2004). Along with segmental duplicates, these several sets of local tandem arrays contribute to the highly homologous sequences of MdPRX gene family.

However, despite their high degree of protein sequence homology, 3D structural predictions revealed substantial variations in the dimensions of the heme-substrate binding interdomain among PRX members, even within the same phylogenetic clades. For instance, although MdPRX90 and MdPRX2 cluster within the same phylogenetic group (Fig. 2, group 1), they exhibit distinct structural characteristics, with MdPRX90 representing a small-pocket PRX variant and MdPRX2 belonging to the large-pocket PRX category. These observations suggest that even minor variations in amino acid sequences can dramatically alter the architecture of the heme-substrate binding domains. Our observations are consistent with the findings of (New, Barsky & Uhde-Stone, 2023), who identified three key amino acid positions (Alpha1, Alpha2, and Beta Buttons) that significantly influence the size of the heme-substrate binding pocket. Specifically, PRXs with smaller heme-substrate pockets tend to feature arginine (R) or lysine (K) at the Alpha1 Button, serine (S) at the Alpha2 Button, and arginine (R) or glutamine (Q) at the Beta Button. In line with this research, we observed a strong correlation between the size of the heme-substrate binding domain and the expression patterns of MdPRXs in WB, MB1, and MB2 (Fig. 7B). MdPRXs belonging to clade 2 of Fig. 7B, such as MdPRX2 and MdPRX76, are exhibit larger heme-substrate binding pockets, whereas those from clade 5, including MdPRX59, MdPRX27, and MdPRX90, display smaller pockets. This clustering pattern further supports the functional relevance of these structural variations in MdPRXs.

Additionally, MdPRX promoters showed cis-regulatory divergence, with segmental duplicates and some tandem arrays having distinct cis-element combinations and motif arrangements. Subcellular localization divergence was further observed between segmentally duplicated pairs and localized tandem arrays, likely resulting from evolutionary diversification of sorting signal peptides. This spatial differentiation implies post-duplication functional specialization, with paralogs gaining distinct sorting signals. As consequence, despite phylogenetic clustering, most MdPRX paralogs exhibited divergent spatiotemporal expression patterns, as evidenced by incongruent profiles in expression heatmaps (Fig. 7). The observed dichotomy suggests that while MdPRXs retained ancestral peroxidase activity (as indicated by phylogenetic conservation), they evolving novel regulatory logics. They have acquired specialized regulatory codes enabling tissue- and stage-specific responses to developmental cues. Thus, the highly homologous and conserved MdPRX proteins response to specific developmental signals in different organs/tissues at different stages. For example, MdPRX13 and MdPRX59 are a pair of segmental duplication genes, with only MdPRX59 exhibiting exclusive expression in vigorous rootstocks. Given the observed cis-regulatory diversification among MdPRX paralogs, future transgenic validation studies should incorporate both functional characterization of promoter architectures and coding sequence analyses. This dual-focused approach is essential for untangling the relative contributions of protein sequence conservation and regulatory element innovation in shaping the spatiotemporal functional diversification of the MdPRX family.

MdPRXs integrate hormone signals and regulate plant development through ROS homeostasis

Since indoleacetic acid (IAA) is one of the substrates of PRX, the relationship between IAA and PRXs has been extensively investigated. However, in this study, we revealed that MdPRXs may also play a significant role in homeostasis, transportation and signal transduction of strigolactone (SL), ABA, and cytokinin (CK), through the processes of reactive oxygen species (ROS) production and scavenging.

The MB apple mutant used in this study is a SL-insensitive mutant. Compared to WT, the IAA content and transcription of the IAA transporter PIN were significantly lower, while the ABA and CK contents were higher in the axillary buds of the MB mutant (Tan et al., 2019). Extensive research has established intricate connections between plant hormone signaling and ROS metabolism. Specifically, apoplastic ROS can be induced and regulated by ABA (Li et al., 2022), while auxins can increase O2•− production and simultaneously induce the expression of reductases such as PRXs, SODs, and CATs, thereby maintaining ROS homeostasis (Mir et al., 2020). This ROS equilibrium reciprocally regulates polar auxin transport dynamics (Cséplő et al., 2021). Additionally, exogenous SL (GR24) treatment has been shown to reduce H2O2 levels in cucumber seedlings (Zhou et al., 2022).

In the tomato system, two NADPH oxidase isoforms, RBOH1 and WFI1, serve as the primary enzymatic sources of apoplastic H2O2 production. The accumulation of H2O2 in the apoplast functions as a critical secondary messenger that activates auxin biosynthesis pathways in the shoot apical meristem, subsequently downregulates CK biosynthesis in stem tissues (Chen et al., 2016). The consequent hormonal imbalance, characterized by elevated auxin-to-cytokinin ratios, effectively suppresses axillary bud development, thereby maintaining apical dominance. Given the dual enzymatic functionality of MdPRXs-functioning either as H2O2-scavenging enzymes that antagonize NADPH oxidase activity or as H2O2-generating enzymes analogous to NADPH oxidases, as previously discussed, we propose that the tissue-specific expression patterns of MdPRXs may mediate the observed physiological changes in WB tissues. Specifically, these expression patterns likely contribute to the downregulation of PIN-formed auxin transporters and the concomitant elevation of cytokinin levels, potentially through modulation of ROS homeostasis and subsequent hormonal signaling pathways. Furthermore, the resulting hormonal imbalance, involving dynamic changes in IAA, CK, and SL levels, appears to establish a feedback regulatory mechanism that modulates MdPRX expression.

Supporting this hypothesis, cis-element analysis revealed the presence of ABA-, IAA-, and SA-responsive elements in the upstream promoter sequences of MdPRXs. These findings strongly suggest the existence of a feedback regulation network between hormonal signaling pathways and peroxidase-mediated ROS homeostasis. Within this regulatory circuitry, PRXs function as molecular hubs that process hormonal signals to finely tune ROS dynamics, thereby regulating lateral shoot differentiation and growth vigor (Fig. 9). According to our proposed regulatory model (Fig. 9), distinct MdPRX members exhibit specific responsiveness to different plant hormones, enabling precise modulation of ROS (H2O2 and O2•−) concentrations within OC and elongation zones of axillary buds. This ROS-mediated regulation subsequently influences multiple physiological processes, including auxin biosynthesis, polar auxin transport through PIN proteins, and cytokinin metabolism. The resulting hormonal crosstalk ultimately determines axillary bud developmental fate as well as growth vigor. However, further studies are needed to validate this hypothesis. For instance, over-expressing specific MdPRXs that are up-regulated in sprouting MB2 buds such as MdPRX90/59/27 could provide critical insights into their functional roles and confirm the proposed regulatory mechanism.

Figure 9 The hypothesized model of MdPRX function in regulating rootstock growth vigor and axillary bud sprouting.

In this model, where PRX acts as a molecular hub, various hormonal signals, including auxin, cytokinins (CK), strigolactones (SL), abscisic acid (ABA), jasmonic acid (JA), and diverse developmental or tissue-specific signals (DS/TSS), interact with the cis-elements of MdPRX genes to activate specific members. Activated MdPRXs generated “large pocket” and “small pocket” variants that shifted ROS balance (H2O2/O2•−). This ROS change both triggered biological effects (e.g., multi-branched or less branched phenotypes, dwarfing or vigorous phenotypes) and provided feedback to regulate hormone synthesis and developmental signaling.

Conclusion

In this study, a comprehensive analysis of the plant-specific CIII PRX gene family in apple was conducted. A total of 99 full-length MdPRX genes were characterized, all of which at least one peroxidase domain. These MdPRXs were unevenly distributed across apple chromosomes, with tandem duplication identified as the primary contributor to the expansion of the MdPRX family. MdPRX genes regulate apple growth and development, as shown by their tissue specific expression patterns. Cis-element, gene expression analyses and protein 3D structure prediction provide insights into the functional roles of MdPRX genes. These findings highlight MdPRX59 and MdPRX99, including their promoters, as key targets for improving dwarf rootstocks through functional validation and breeding. Additionally, the over-expression of MdPRX59, MdPRX90, and MdPRX27 presents a promising strategy to achieve a more-branching phenotype.

Supplemental Information

Supplemental Information 1 MIQE checklist

Supplemental Information 2 qRT-PCR data

Supplemental Information 3 The MIQE of qRT-PCR

Supplemental Information 4 MdPRX coding (CDS) sequences

Supplemental Information 5 Ka_Ks values and divergence time of MdPRX collinear pairs

Supplemental Information 6 The predicted MdPRX protein parameters and subcellular localization

Supplemental Information 7 The predicted MdPRX signal peptide by TARGETP

Supplemental Information 8 Batch Batch SMART information of MdPRXs

Supplemental Information 9 The annotation revision of MdPRX61

(a) the original annotation of MD11G106900 which predicted to be MdPRX61, it has 19 exons but there is no peroxidase domain in the first 15 exons (b) the RNA-Seq read mapping to mRNA MD11G106900 showed that the first 15 exons were expressed independently (c) the revised annotation of MdPRX61 by dividing the original MD11G106900 into two separate transcripts, the last four exons (16-19 exons) which contain a signal peptide and a peroxidase domains were renamed as MdPRX61.

Supplemental Information 10 The 1-10 conserved motif logo of MdPRXs

Supplemental Information 11 Phenotypes of WB and WT and sampling location of RNA-Seq and qRT-PCR

(A)Axillary bud phenotypes from the shoot apex to the branches in WT and MB at 60 DAB. The WB,MB1-MB3 indicates axillary buds and differentiated elongation zone of new branches used for RNA-seq and qRT-PCR. Scale bar = 5.0 mm. (B) Branching phenotypes of the WT and MB. Branching phenotype of the primary shoot of the WT and MB at 60 DAB. White arrows indicate the axillary buds or branches of partial nodes. #1 and #2 indicate different lines.

Supplemental Information 12 Primer sequences

Supplemental Information 13 The statistical data of MdPRX cis-elements

Additional Information and Declarations

Competing Interests

Author Contributions

Data Availability

The authors declare there are no competing interests.

Yao Lu performed the experiments, analyzed the data, prepared figures and/or tables, authored or reviewed drafts of the article, and approved the final draft.

Rongqun Ma conceived and designed the experiments, prepared figures and/or tables, and approved the final draft.

Kunhao Wu analyzed the data, prepared figures and/or tables, and approved the final draft.

Jilu Sun analyzed the data, prepared figures and/or tables, and approved the final draft.

Yutong Li analyzed the data, prepared figures and/or tables, and approved the final draft.

Jiawei Zhao analyzed the data, prepared figures and/or tables, and approved the final draft.

Zhenbao Qi analyzed the data, prepared figures and/or tables, and approved the final draft.

Guangli Sha analyzed the data, authored or reviewed drafts of the article, and approved the final draft.

Hongjuan Ge conceived and designed the experiments, authored or reviewed drafts of the article, and approved the final draft.

Yanjing Shi conceived and designed the experiments, performed the experiments, authored or reviewed drafts of the article, and approved the final draft.

The following information was supplied regarding data availability:

The raw sequencing data generated in this study are available at NCBI:

PRJNA826123, PRJNA801073, PRJNA308148; and GEO:

GSE274104, GSE253335, GSE276181; SRR309569.

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
