# Peer review of "Genomewide analysis of the Class III peroxidase gene family in apple (Malus domestica)"

_PeerJ, doi:10.7717/peerj.19741_

## Round 0.1 · original submission · Minor Revisions

The overall MS is well-written and technically sound. There are some minor errors that needs to be addressed before publication as follows:

--MS must be checked for language-related issues and grammatical/typographic errors.
--In the Methods section, provide additional details and justifications about RNA seq and experimental data usage.
--Details figure sections are essential.

·

Basic reporting

The authors have made a significant effort to annotate and analyze the class III peroxidase (PRX) genes in the Malus domestica genome. They provide valuable insights into the structure, function, and potential roles of these genes in apple. The study is relevant to the current field of plant genomics, especially in the context of understanding gene families involved in stress responses and developmental processes. The manuscript is well-structured, with a clear hypothesis and a thorough exploration of the available data.

However, I noticed a few minor spelling errors and inconsistencies in the file naming scheme (e.g., in Additional Files 2 and 3) that need to be corrected.

In Line 246, the authors state that 14 PRX gene members are predicted to be mitochondria-localized, but Additional File 5 only supports 3 of these genes being classified as “mito.” This discrepancy should be addressed and clarified, either by revising the prediction criteria or explaining the difference between the predictions in the text and the supporting data.

The quality of Figure S2 could be improved. Currently, it is somewhat difficult to interpret due to the resolution and clarity of the labels. Enhancing the quality of this figure would help the readers better grasp the key findings.

Figure 7 also suffers from similar issues. It could benefit from a clearer layout and more legible labeling to ensure that the data presented are easily interpretable.

Experimental design

The authors mention that signal peptides were not found in the majority of the PRX genes. While this observation is valuable, it would be beneficial to have more confidence in these results by using specialized tools like SignalP or TargetP, which are more reliable for predicting signal peptides. The authors should consider running these tools to validate their results and provide further evidence for their claim.

Validity of the findings

Since AlphaFold2 structures are publicly available for many proteins, including peroxidases, the authors should consider using this tool for further structural analysis of the PRX genes. Structural predictions could provide insights into the structure-function relationship of these genes, which would significantly enrich the manuscript and make it more comprehensive. The addition of such structural data could provide functional insights, especially in relation to enzymatic activity or interactions with substrates.

Additional comments

This manuscript presents a valuable contribution to the understanding of the class III PRX gene family in Malus domestica. The research question is well-defined, and the study is based on a sound methodology. However, addressing the points mentioned above will enhance the overall quality of the manuscript and increase its impact on the scientific community. The authors are encouraged to make these revisions and resubmit the paper for further consideration.

·

Basic reporting

The abstract provides a comprehensive summary of the study, but it lacks emphasis on the novelty and specific contributions of the research. It is recommended to highlight the unique aspects of this study compared to existing research on Class III PRXs and clarify how the findings advance apple crop improvement efforts beyond prior knowledge.

The manuscript is generally well-written but contains minor grammatical errors and overly long sentences. Simplifying complex sentences for better readability and ensuring consistent usage of technical terms (e.g., “MdPRX” vs. “PRX genes”) would enhance clarity and coherence.

The references are comprehensive but lack consistency in formatting. Ensuring that all references follow the same style (e.g., journal names abbreviated or not) would improve the overall presentation. It is also important to verify that all cited works are relevant and up-to-date.

Experimental design

The introduction offers relevant background information on PRXs, but the focus on the research gap could be sharper. Explicit identification of knowledge gaps, particularly in relation to PRXs in Malus domestica, would improve this section. Additionally, elaborating on the significance of this research for the apple industry, such as the economic implications of growth vigor and branching, would strengthen its impact. Overly detailed descriptions of ROS and PRX functionality could be condensed or shifted to the discussion section.

The methods section is well-structured but could benefit from additional clarity in certain experimental procedures and data analysis approaches. More details should be provided about the criteria used for identifying and excluding candidate PRX genes during annotation. Additionally, the normalization techniques for RNA-Seq data and statistical adjustments, if any, should be explained. The primer design and validation processes could be elaborated with citations for the methodologies used. Lastly, a brief justification for using specific RNA-Seq datasets (e.g., GSE253335, GSE274104) would enhance the transparency of the study.

The results section covers a wide range of findings but could be presented with better visualization and interpretation. For the identification of PRX proteins, including a summary table of MdPRX genes with key attributes like chromosomal location and domain structure would aid understanding. The evolutionary insights provided in the phylogenetic analysis are intriguing but need further interpretation regarding the functional implications of observed duplications. The spatio-temporal expression patterns, while comprehensive, lack detailed biological interpretation of the data. Additionally, the discussion on the role of MdPRXs in rootstock-scion interactions, though novel, requires clearer articulation of how specific expression profiles correlate with scion performance or dwarfing ability.

Validity of the findings

The figures and tables are relevant but could be made more self-explanatory. Ensuring that all figures have detailed captions explaining the methods and significance of the data presented is essential. Adding schematic diagrams to illustrate key findings, such as PRX gene duplication events, would enhance the visual impact and accessibility of the results.

The discussion is thorough but can be more concise and better aligned with the research objectives. Repetition of background information already presented in the introduction should be reduced. Expanding on the broader implications of the findings for breeding strategies and practical horticulture would add value. Future research directions, such as functional validation of candidate MdPRX genes, should also be elaborated.

The conclusion effectively summarizes the findings but could better emphasize their practical applications. Highlighting specific recommendations for apple breeding programs and addressing how this research lays the groundwork for genetic manipulation or molecular breeding efforts would strengthen the conclusion.

Additional comments

The manuscript would benefit from revisions to enhance its clarity, coherence, and impact. The abstract should emphasize the novelty and practical implications of the research, while the introduction needs a sharper focus on the research gaps and objectives. The methods section requires additional details and justifications, particularly regarding RNA-Seq data usage and experimental criteria. In the results section, better visualization and more thorough interpretation of phylogenetic and tissue-specific expression patterns are needed. The discussion should be streamlined to reduce repetition and focus more on practical applications and implications for breeding strategies. Figures and tables require improved captions and supplementary visual aids to enhance accessibility. Lastly, consistent reference formatting and minor language corrections would further improve the manuscript.

---

## Round 0.2 · Minor Revisions

Authors have addressed most of the comments. However, still some minor comments need to be incorporated. Revise the manuscript as per reviewers' suggestions and proofread the whole manuscript for grammatical and typographical mistakes.

·

Basic reporting

The authors have made a commendable effort to revise the manuscript, addressing the suggestions raised in the previous round. They have carefully corrected spelling errors and updated the file naming scheme. Additionally, they have improved the quality of the figures compared to the earlier version. I am satisfied with the revised manuscript and support proceeding to the next steps.

Experimental design

The authors have included the evidence of TargetP and Wolf PSORT predictions to validate the presence of signal peptides on MdPRX genes.

Validity of the findings

The authors conducted a structural analysis of the MdPRX genes and validated variations in their substrate pockets. This analysis was essential, as it enabled them to infer functional differences among the otherwise similar-looking structures.

·

Basic reporting

The revised manuscript titled "Genomewide analysis of the Class III peroxidase gene family in apple (Malus domestica)" presents a comprehensive and methodologically rigorous genome-wide investigation of the Class III peroxidase (PRX) gene family in apple. It makes a significant contribution to plant molecular biology by linking PRX gene family expansion and functional divergence with ROS-mediated developmental regulation, particularly in rootstock vigor and axillary bud differentiation. The integration of multi-omics approaches—phylogenetic analysis, structural modeling, transcriptome profiling, and expression validation—is highly commendable and adds substantial depth to the findings.

Experimental design

The authors employ a robust set of bioinformatics and molecular tools, including HMMER for gene identification, MEME for motif analysis, MCScanX for duplication analysis, and Swiss-Model/AlphaFold for protein structure prediction. The use of multiple RNA-seq datasets across different tissues and developmental stages significantly enhances the resolution of expression dynamics. Additionally, qRT-PCR validation supports the transcriptomic data. However, more detailed reporting of qRT-PCR primer efficiency, product specificity (e.g., melting curves), and amplification conditions would further strengthen reproducibility.

Validity of the findings

The interpretation of expression patterns in light of structural modeling of PRXs (specifically pocket size variability) is compelling. The authors establish a plausible link between PRX structure, ROS activity, and developmental regulation. Nonetheless, further quantitative descriptors of the interdomain pocket size would enhance the precision of this classification into "small-pocket" and "large-pocket" PRXs. The proposed functional dichotomy—PRXs as ROS generators versus scavengers—is persuasive but could benefit from further validation, potentially through functional assays or enzymatic activity measurements.
The discussion effectively integrates current knowledge on hormone signaling (auxin, cytokinin, ABA, strigolactone) and ROS homeostasis, positioning PRXs as key molecular hubs in hormonal feedback circuits. The manuscript suggests that PRX expression is both regulated by and contributes to hormonal balance in the shoot apical meristem and rootstock tissues. This conceptual model is compelling, yet would benefit from a visual diagram summarizing the feedback loops proposed.

Additional comments

The authors have clearly made significant efforts to address prior reviewer feedback. They have expanded the functional discussion, clarified methodological aspects, and introduced structural-functional analyses that significantly elevate the depth of the study. These revisions are appreciated and demonstrate the authors’ commitment to improving the scientific quality and impact of the work.

The manuscript is generally well-written and logically organized, with a clear progression from gene family identification to expression profiling and functional inference. However, minor grammatical issues and typographical inconsistencies (e.g., spacing, inconsistent gene naming conventions) should be addressed. Streamlining technical descriptions in the methods section could improve accessibility for a broader readership. A graphical summary of key findings (e.g., a conceptual model of PRX function) would greatly enhance reader comprehension.

This manuscript presents original and important findings supported by rigorous analysis. It offers valuable insight into PRX gene family function in apple development and has clear translational potential for rootstock improvement. I recommend acceptance after minor revision to enhance language consistency and correct typographical errors.

---

## Round 0.3 · accepted · Accept

The authors have addressed all the comments. The manuscript can be accepted for publication.

·

Basic reporting

no comments

Experimental design

no comments

Validity of the findings

no comments